# Long-Term Annual Surface Water Change in the Brazilian Amazon Biome: Potential Links with Deforestation, Infrastructure Development and Climate Change

**Carlos M. Souza, Jr. [1],\*, Frederic T. Kirchhoff [1], Bernardo C. Oliveira [2] , Júlia G. Ribeiro [1] and Márcio H. Sales [3]**

[1] Instituto do Homem e Meio Ambiente da Amazônia (Imazon), Belém 66055-200, Brazil;
frederic@imazon.org.br (F.T.K.); juliagabriela@imazon.org.br (J.G.R.)

[2] Science Program, WWF-Brasil, Brasília 700377-540, Brazil; bernardo@wwf.org.br

[3] MHR Sales Consultoria, Belém 66633-090, Brazil; marciosales@outlook.com

\* Correspondence: souzajr@imazon.org.br; Tel.: +55-91-3182-4000

**Abstract:** The Brazilian Amazon is one of the areas on the planet with the fastest changes in forest cover due to deforestation associated with agricultural expansion and infrastructure development. These drivers of change, directly and indirectly, affect the water ecosystem. In this study, we present a long-term spatiotemporal analysis of surface water annual change and address potential connections with deforestation, infrastructure expansion and climate change in this region. To do that, we used the Landsat Data Archive (LDA), and Earth Engine cloud computing platform, to map and analyze annual water changes between 1985 and 2017. We detected and estimated the extent of surface water using a novel sub-pixel classifier based on spectral mixture analysis, followed by a post-classification segmentation approach to isolate and classify surface water in natural and anthropic water bodies. Furthermore, we combined these results with deforestation and infrastructure development maps of roads, hydroelectric dams to quantify surface water changes linked with them. Our results showed that deforestation dramatically disrupts small streams, new hydroelectric dams inundated landmass after 2010 and that there is an overall trend of reducing surface water in the Amazon Biome and watershed scales, suggesting a potential connection to more recent extreme droughts in the 2010s.

**Keywords:** Amazon Biome; land cover change; climate change; deforestation; infrastructure development

## 1. Introduction

The water ecosystem in the Brazilian Amazon Biome (hereinafter Amazon) is under pressure from deforestation, land use activities, urbanization, road expansion and by the construction of hydroelectric dams [1–3]. These anthropogenic changes have accelerated in the past 50 years in this region, causing negative impacts on aquatic ecosystems. For example, stream flows are disrupted isolating fish communities and preventing the local population accessing water resources [4]. Water pollution, mostly from agriculture [5] and gold mining activities [6,7], also changes aquatic biodiversity, chemistry and sediment discharge rates. The construction of dams affects natural surface water flow, river connectivity and aquatic biodiversity migration [8–10]. Large-scale deforestation can also disrupt biogeochemical and hydrological cycles affecting the amount and volume of water [11]. The Amazon forests, for example, control regional rainfall and recycle at least 50% of the water supplying all the aquatic ecosystems in the Basin [12].

The water ecosystems in the Amazon are also under the influence of climate change. There is strong scientific evidence of more frequent, intense, more prolonged and extreme droughts and floods in the Amazon [13,14], which affects the hydrological cycle directly within and outside the region. In 2010, the combined effect of severe El Niño and the warming of the North Atlantic may be the cause of the lowering of the Amazon River to the lowest level registered in modern history [15]. This severe drought has affected the major tributaries of the Amazon River, floodplain communities, cities and villages that depend on rivers for food, water consumption, and transportation. Therefore, the combined pressure of anthropogenic interferences and climate change can increase the vulnerability of the Amazon water ecosystem [2].

Remote sensing data is being considered a primary source of information to assess land cover change in this region [16], with several applications being developed to monitor deforestation (e.g., [17–19]). Satellite measurements are also a primary source of information to directly enable the mapping of surface water of the aquatic ecosystem in floodplains, rivers, channels, lakes and reservoirs [20–22]. Due to the low density of water gauges in the Amazon region, satellite imagery is also the only source of data for reconstructing long time-series (i.e., +30 years) of the spatial-temporal dynamics of surface water in this region [21]. Examples of surface water mapping obtained with remote sensing include the Global Surface Water (GSW) study based on a 30-year analysis of Landsat images at the pixel level [23,24], and a novel surface water sub-pixel classifier (SWSC) algorithm [25]. GSW provides a valuable dataset for characterizing spatial and temporal surface water dynamics, but the SWSC outperformed GSW by revealing undetected surface water features in wetlands, floodplains, small rivers, streams and lakes because it reveals more mixed water with vegetation and soil land cover [25]. The advantage of GSW is that its dataset includes intra-annual surface water changes on a monthly basis, while SWSC mapped surface water annually in the dry season (circa June through October). However, wall-to-wall mapping of the Amazon biome on a monthly basis is compromised by high cloud frequency in the rainy season, making the dry season (i.e., June through October) the more likely period of the year for acquiring less cloudy Landsat imagery [26].

In this study, we compiled 33 years of Landsat imagery to generate annual surface water mapping of the Amazon Biome using the novel SWSC method, focusing on the mapping of surface water in the dry season. Next, we assessed annual land to water and water to land changes from 1985 to 2017. Finally, we used existing annual deforestation maps from PRODES, the Brazilian Amazon satellite monitoring system [27] and hydroelectric dams to evaluate the impact of these vectors in the water ecosystem. These analyses allowed us to estimate the annual extent variability of surface water in the biome, and the impact of deforestation and hydroelectric dams on the water ecosystem. We conducted the remote sensing and the spatial analyses in the Google Earth Engine platform [28], which we describe in the method section together with an overview of the study area, the datasets used and the remote sensing and spatial analyses techniques. In the results and discussion section, we present the annual time-series results of surface water mapping and the interchange between water and landmass dynamics, followed by the impact of deforestation and infrastructure impact assessment on surface water at the Amazon biome and watershed scales, and discuss the main findings of these topics. We then draw our conclusions and next steps in the final section.

## 2. Materials and Methods

### 2.1. Study Area

The study area covers the Amazon Biome (hereafter Amazon), with an area of 4.2 million km$^2$ mostly comprised of evergreen forests, but also including natural grasslands, wetlands, and regions converted to agriculture and cattle ranching (Figure 1). The Amazon region is being changed rapidly due to deforestation [17,29], forest degradation rates [30], mining activities [31] and infrastructure development, including the construction of roads and hydropower dams [32–34]. These types of changes lead to the disruption in water flow, new permanently inundated areas and water

contamination [35,36]. This region possesses 40% of the living species on the planet [35,37], and owns the world's largest freshwater reserves [38].

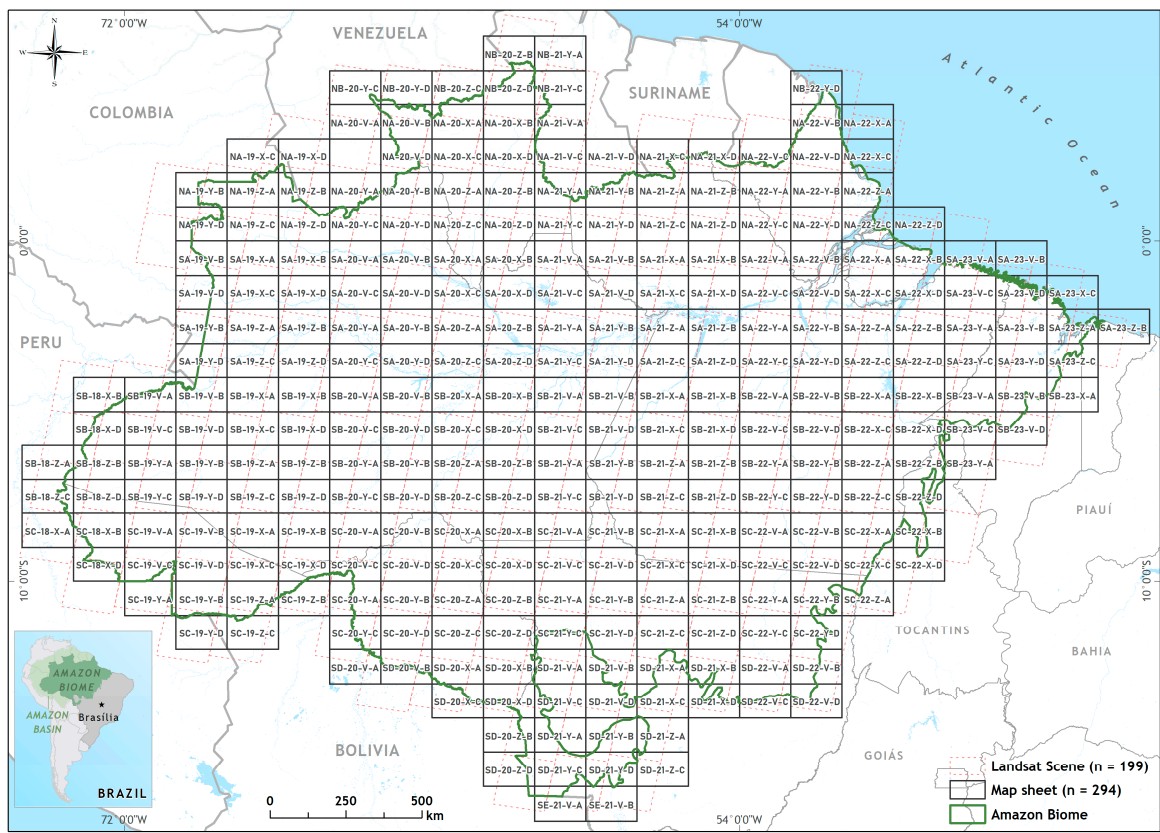

**Figure 1.** Amazon biome study area, Landsat scenes (*n* = 194) and the map sheets (*n* = 294) used for producing annual mosaics of Landsat to map annually surface water from 1985 to 2017.

### 2.2. Landsat Dataset

We have analyzed Landsat Data Archive (LDA) [39] available in the Earth Engine, covering the period of 1985 to 2017, to map and analyze the dynamics of surface water. A total of 194 Landsat scenes cover the study area (Figure 1). This tropical region is subject to persistent and extensive cloud cover [26]. Due to this cloudy condition, which blocks Landsat ground observation, we have produced an annual mosaic to remove clouds using a median statistical filter available in the Earth Engine. The mosaic area covers the tile boundary of the International Map of the World to the Millionth on the scale of 1:250,000, comprising 1°30′ of longitude by 1° of latitude (Figure 1). Each mosaic requires two to four Landsat scenes and a total of 294 map mosaics are necessary to cover the Amazon Biome boundary. We produced the map sheet annual mosaics with L1T Surface Reflectance for Landsat 5 (USGS, Sioux Falls, SD, U.S.) (1885–2002), Landsat 7 (USGS, Sioux Falls, SD, U.S.) (2000–2015) and Landsat 8 (USGS, Sioux Falls, SD, U.S.) (2013–2017), pre-processed by USSG. The annual timeframe for image acquisition concentrated mostly between June 1st and October 31st—the months most likely to allow ground measurements from the Landsat sensors [26].

### 2.3. Image Processing

We selected and processed the Landsat images directly in the Earth Engine Platform [28]. The image processing steps for mapping surface water are summarized in Figure 2 and presented in detail below.

Step 1: Build the annual Landsat mosaic

We searched and filtered the ortho-rectified L1T (Level 1 Terrain) Landsat collections based on sensor type, year of acquisition, the time period of the year (i.e., June 1st through October 31st) and cloud cover up to 30%. This procedure results in a list of Landsat scenes for each map sheet for each year from 1985 to 2015. Next, we applied the pixel quality band of the Landsat sensors to map and mask out clouds. A median statistical reducer was then used to estimate the best pixel observation for the year. We also removed the edges of the Landsat scenes, prior to the application of the median filter, using a 500 m buffer to avoid the inclusion of spurious data. Cloudy pixels not detected by the pixel quality cloud mask were removed using the Cloud fraction ($\geq$10%; see Step 2: Spectral Mixture Analysis (SMA)). This process resulted in an annual mosaic of Landsat images with surface spectral information for the period selected (i.e., June 1st through October 31st) (Figure 2).

Step 2: Spectral Mixture Analysis (SMA)

SMA is a well-established image-processing tool for estimating the sub-pixel composition of Landsat pixels [40], and applications have been proposed for mapping surface water bodies and wetlands [40,41]. Here, we used annual median surface reflectance mosaics, with Landsat bands 1–5 and 7 obtained in Step 1, to estimate the Landsat pixel composition. The SMA model was used to calculate the proportion of purer spectral signature (named endmembers) of Green Vegetation (GV), Soil, Non-Photosynthetic Vegetation (NPV) and Cloud in each pixel, based on a generic set of endmembers previously defined [42]. The SMA procedure was implemented in the Earth Engine using the available unmixing algorithm, resulting in fractional or composition images of each endmember. We calculated the shade fraction by subtracting the sum of all endmember fractions from 1 (i.e., 100%). Therefore, the sum of all endmember fractions equals 1. Details of the SMA model and endmembers used in this study can be found elsewhere [17,42].

Step 3: Surface Water Sub-Pixel Classification (SWSC)

The fraction images obtained in Step 3 through the SMA model were used as inputs for Surface Water Sub-Pixel Classifier (SWSC) to generate surface water maps. Surface water pixels have a high fractional Shade value (i.e., >65%). The Soil fraction can increase in water with high content of sediments (i.e., Soil), and sandbanks or bare ground along the border of rivers, lakes and wetlands. GV composition is also expected to increase in floodplains, in wetlands and along the margins of rivers and lakes. The SWSC uses simple binary decision rules to classify pixels (not blocked by clouds) as surface water and no surface water, as follows (Figure 2):

*Surface Water: Shade $\geq$ 65% and (Soil + Vegetation) $\leq$ 20%.*
*Land: otherwise*

Pixels with more than 10% of Cloud fraction were masked out to remove remaining cloudy pixels not detected by the forest masking procedure described above. These simple binary rules were applied to all fractional images of each annual mosaic with a threshold value for Shade varying slightly (i.e., 1%–5%), across space and time due to changes in illumination in the Landsat scenes (Figure 2).

Step 4: Post-Classification

We applied three post-classification procedures to the annual surface water maps obtained with the SWSC. First, we used spatial filtering to remove spurious classification results. This spatial filter uses a region-labeling algorithm available in the Earth Engine to identify each contiguous surface water bodies or objects. Surface water objects with less than three pixels were removed from each annual map of surface water because they were too small to be classified as water bodies. We then applied a temporal moving window filter with a kernel size of three, to reclassify Cloud if the previous and the subsequent years belong to the same class (i.e., Land or Surface Water). Finally, we also produced a Water Occurrence map as proposed by [23] to characterize and quantify the

frequency of Surface Water mapping between 1985 and 2017. Next, we reclassified the pixels that were classified less than 10% of the time (i.e., ~3 years) as surface water over the 33 years as Land surface. This procedure allowed removal of misclassification of surface water associated with burned areas and high shade/shadow in urban areas.

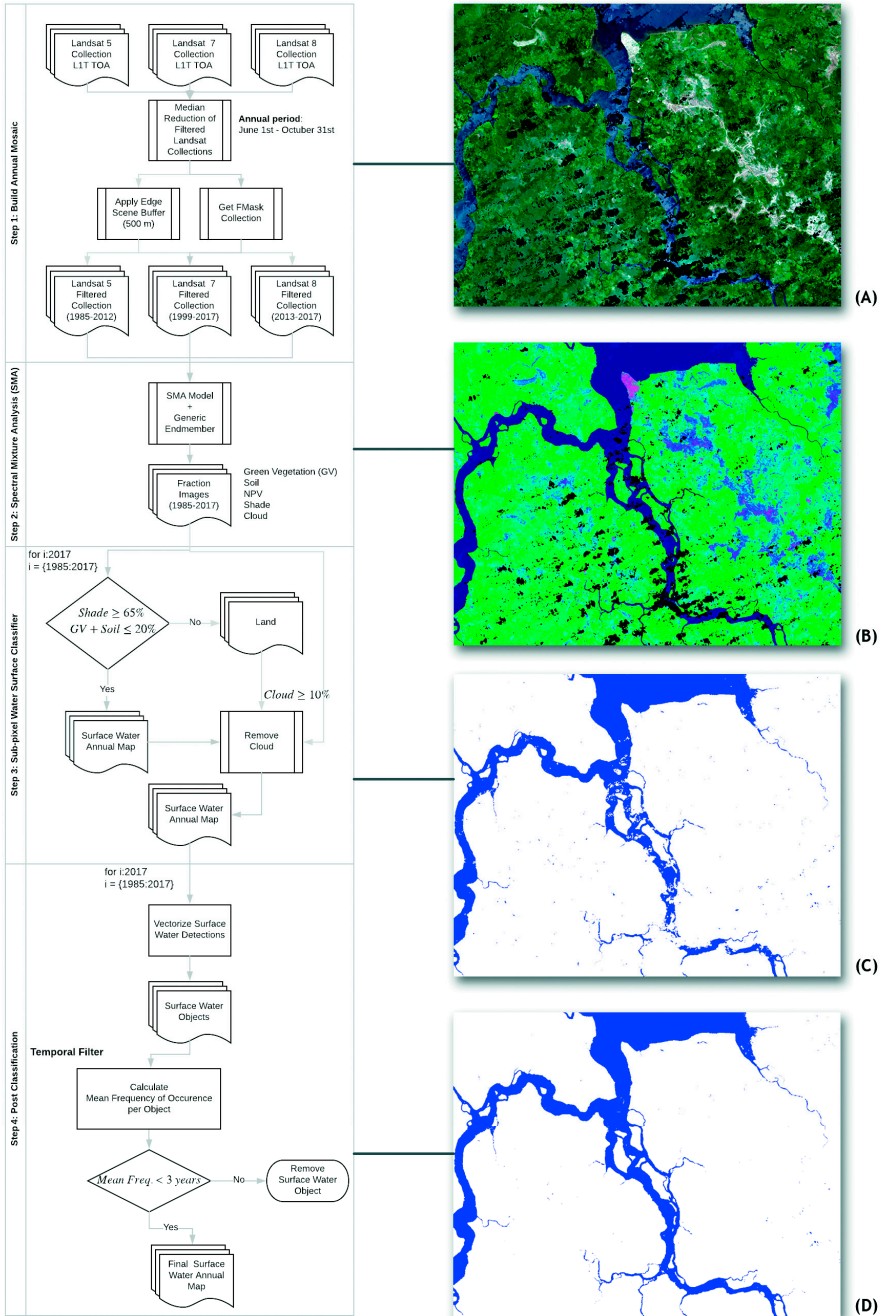

**Figure 2.** Image processing applied to annual Landsat datasets acquired between June and October to derive surface water maps, (**A**) annual median mosaic (**B**) SMA color composite (Red—Soil, Green—Green Vegetation, Blue—Shade), (**C**) surface water classification, and (**D**) temporally filtered surface water.

## 2.4. Surface Water Object Classification

Isolating surface water objects was necessary to classify them into natural and artificial surface water bodies. For that, we applied the region labeling algorithm available in the Google Earth Engine,

first to isolate the surface water objects and assign them a unique identifier or label. Next, we calculated the area of each water object and extracted their morphological attributes including area, perimeter, area-perimeter ratio and convexity/concavity degree. The area and morphological attributes of the water bodies were used as data features to train and run a random forest classifier (with 100 trees) to classify the following surface water classes:

Natural: rivers, lakes, wetlands.
Anthropic: hydroelectric dams, small dams, and gold mining.

*2.5. Characterization of Surface Water Dynamics*

We evaluated surface water dynamics at the Amazon Biome (Figure 1) and watershed scales. To do that, we measured annual variability of surface water extent mapped with the SWSC, and the changes from Land to Water (i.e., gain) and Water to Land (loss). The watershed analysis allowed us to account for regional variations in gains and losses of surface water during the timeframe of this study (1985–2017). We also quantified the extent of annual surface water in each watershed. We used level 4 watershed boundary [43] made available the federal hydrographical institution of Brazil (i.e., ANA—*Agência Nacional de Águas*). Furthermore, we calculated for each watershed the mean surface water area from 1985 to 2017, and estimated the percent change from the mean for each watershed to assess which watersheds had gains and losses in surface water in 2017.

*2.6. Deforestation Impact on Surface Water*

We combined spatially explicit deforestation data from Prodes, the Brazilian Amazon Forest Monitoring Program, with surface water derived from SWSC and the types of water body objects (i.e., Natural and Anthropic). This spatial analysis covered a shorter period (i.e., 2000–2017) because Prodes annual digital maps are available only after 2000. Next, we analyzed the amount and size of artificial yearly mapped water bodies (hydroelectric dams, small dams, and gold mining) inside the annually deforested areas. This deforestation analysis was also conducted at the watershed scale to assess correlation with surface water extent. For that, we analyzed the effect of deforestation and forest cover in surface water area using a linear mixed model applied to classes of watersheds that gained and lost surface water (see Section 2 of Supplementary Materials).

*2.7. Accuracy Assessment*

We estimated the accuracy statistics for Land and Water classes for each year from 2000 to 2015 using 1000 stratified random sample points. We used Landsat color composites and asked three independent image analysts to classify the sample points based on visual interpretation of three classes: Water, Land or Cloud. Next, we generated a final reference dataset combining the results from the three analysts results using a majority rule, i.e., assign the final class when more than two analysts agreed (see [44] for more details) and removing sample points classified as Cloud. The average number of total points evaluated per year was 838 (s = 75) after removal of unobserved pixels, with 93 (s = 6) for Water, and 745 (s = 78) for Land. Accuracy statistics were calculated using area-adjusted corrections, proposed as a best-practice procedure by the literature [45].

## 3. Results and Discussion

*3.1. Surface Water Classification*

The SWSC algorithm applied to SMA fractions derived a long-term (i.e., 33 years) dataset of annual surface water for the Amazon Biome. This annual mapping focused on the dry season (circa June 1st, through October 31st), which increases the likelihood of acquiring Landsat images in the Amazon region with less cloud [26], allowing annual map a large extent of the study area. Due to a lower precipitation regime during this period of the year [43], the surface water detected is expected to

capture the lowest water level of rivers, lakes and flood plains (although torrential rainfall can generate localized flooding during the dry season). We also processed the GSW dataset and generated monthly statistics of the surface water extent, which revealed that GSW mapped less surface water during the rainy season relative to the dry season (Figure S3). This suggests that the optimal period to monitor surface water with optical Landsat data is between June and October, as applied in this study.

The Landsat median-reduced mosaic for 2017 generated normalized SMA fraction images over the 33-year timespan of this study. Figure 3 shows a subset of this dataset as an example of the image processing techniques applied to the Landsat dataset. Surface water bodies show up as dark blue colors in SMA fraction color composites of Soil (Red), Green Vegetation (Green) and Shade (Blue) providing high contrast with land cover categories (i.e., forest, agriculture and urban lands) (Figure 3A). The application of the binary hierarchical rules of the SWSC algorithm to the Landsat time-series fractions produced annual maps of Surface Water and Land (Figure 3B). Combining these annual surface water maps allowed us to generate frequency or water occurrence maps of surface water between 1985 and 2017 (Figure 3C). The average user's accuracy of the SWSC algorithm through 2000 to 2015 was 95% (s = 1), with 84% (s = 2) average user accuracy for the Water class, and 96% (s = 1) for the Land.

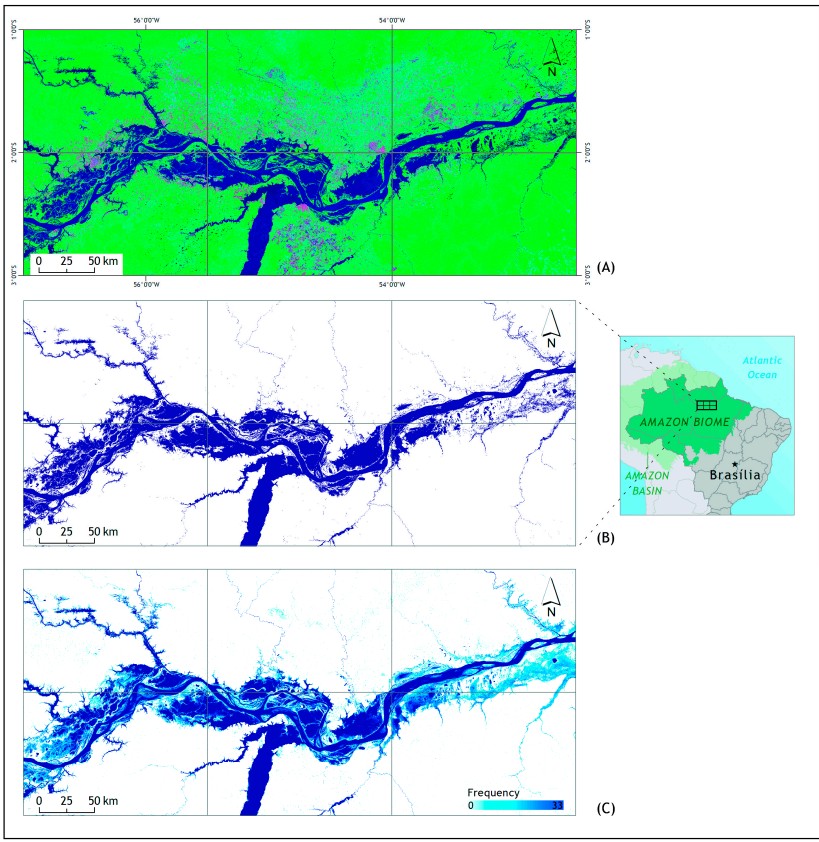

**Figure 3.** Example of Spectral Mixture Analysis (SMA) results for a portion of the Amazonas River and Tapajós tributary. In (**A**), a color composite obtained by combining fraction images of Soil (Red), Green Vegetation (Green) and Shade (Blue). In (**B**), the water classification based on the surface water sub-pixel classifier (SWSC) and in (**C**) the water frequency map over 33 years. Light blue areas adjacent to dark blue ones are likely to represent land areas susceptible to flooding, and bright blue areas can also designate areas that had the fewer cloud-free satellite observation such as in the left side of the Figure 3C.

We estimated the annual surface water area from the SWSC maps (Figure 4). The maximum surface water mapped was in 1991 covering an area of almost 140,000 km$^2$ and the minimum one was in 2016 (i.e., 108,674 km$^2$), which is considered an extremely dry year in the Amazon region [44].

The range in surface water between 1991 and 2016 was 30,838 km$^2$. However, surface water mapped in 2016 is also 16,180 km$^2$ below the average surface water mapped from 1985 to 2017 (i.e., 33 years) (Figure 4A). The second lowest surface water mapped was in 1985 (112,599 km$^2$/year). Surface water extent also varied throughout the decades.

We also estimated the minimum, maximum, mean and the standard deviation (s) of surface water for circa decade periods (Figure 4B, Table 1). The average surface water between 1985 and 1989 was 123,274 km$^2$/year (s = 5997). The largest surface water extent mapped was in the 1990s with an average of 130,379 km$^2$/year (s = 5302), followed by a decrease in surface water in the early 2000s and an increase in 2005. The average surface water mapped during the 2000s was 127,265 km$^2$/year (s = 4593), which is slightly lower than the average for the 1990s (Figure 4B). However, after 2010, a constant annual decrease in surface water was detected (Figure 4A). Between 2010 and 2017, the SWCS detected the lowest average of surface water in the Amazon Biome relative to the past three decades, falling to 116,811 km$^2$/year (s = 5702) (Figure 4B Table 1).

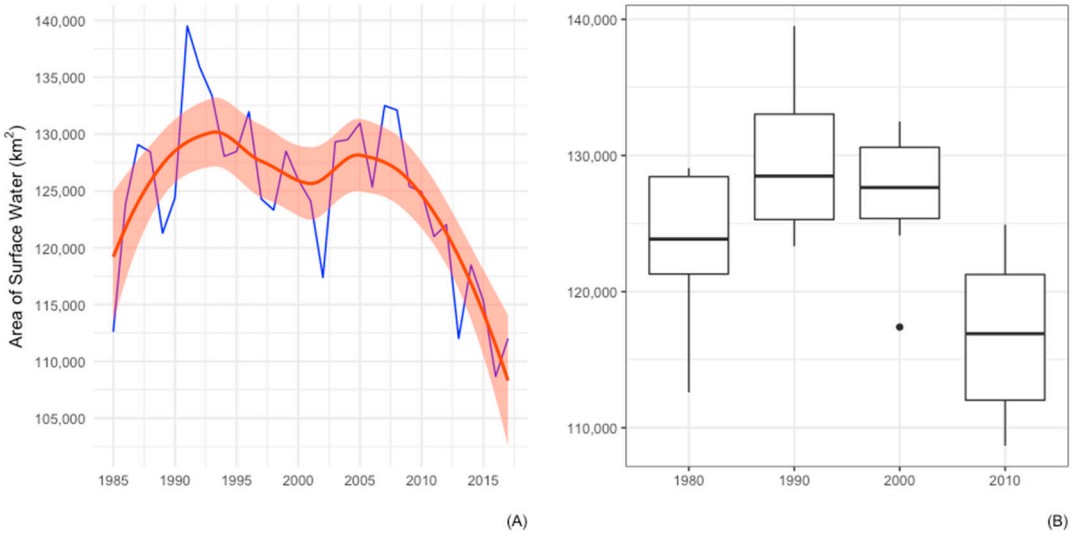

**Figure 4.** Annual surface water extent mapped with SWSC with Loess trend smooth pattern (red line) and 95% confidence interval (**A**). Box plot statistics are presented in (**B**) for circa decade periods.

**Table 1.** Surface water mapping summary statistics estimated for decade periods.

| Period (# Years) | Minimum | Maximum | Mean | Standard Deviation (s) |
|---|---|---|---|---|
| | (km$^2$) | | (km$^2$/year) | |
| 1985–1989 | 112,599 | 129,072 | 123,274 | 5997 |
| 1990–1999 | 123,325 | 139,512 | 130,379 | 5302 |
| 2000–2009 | 117,396 | 132,498 | 127,265 | 4593 |
| 2010–2017 | 108,674 | 124,923 | 116,811 | 5702 |

*3.2. Surface Water Dynamics*

Investigating annual surface water change can help to explain the overall trend in the decreasing yearly surface water extent detected during the dry season of the Amazon region. The most significant reduction in surface water happened in the 2010s with almost 13,000 km$^2$ of difference in surface water between 2010 and 2017 (i.e., ~10%; Figure 4). A 10% reduction in average surface water extent between the 1990s and the 2010s was also observed (Table 1), but a constant decrease in the surface water was the overall spatiotemporal pattern after 2010.

We characterized the surface water dynamics by decoupling annual changes between Water to Land and Land to Water from areas that remained as water (Figure 5). The results of the decoupling analysis of surface water change showed that areas mapped in the Remains Water class did not

show the overall trend of decreasing surface water over the period investigated (i.e., 1985 through 2017). The average surface area in this class of water was around 100,000 km$^2$/year (i.e., 2.4% of the Amazon Biome). However, areas that changed from Land to Water, and from Water to Land showed a decreasing trend in surface water extent (Figure 5B,C) as detected in the total surface water mapped annually (Figure 3A). The rate of change over 33 years obtained with a linear regression model showed a decrease in surface water extent of 350 km$^2$/year over the 33-year timespan in the areas that underwent an interchange between landmass and water. We observed that between 1985 and 2000 these annual changes were not pronounced, but after that point in time, we detected a constant decrease in the extent of Land to Water and Water to Land changes (Figure 5B,C). However, the fastest shrinkage of the areas subjected to this type of dynamics happened between 2010 and 2017, with an average decrease of nearly 1400 km$^2$/year.

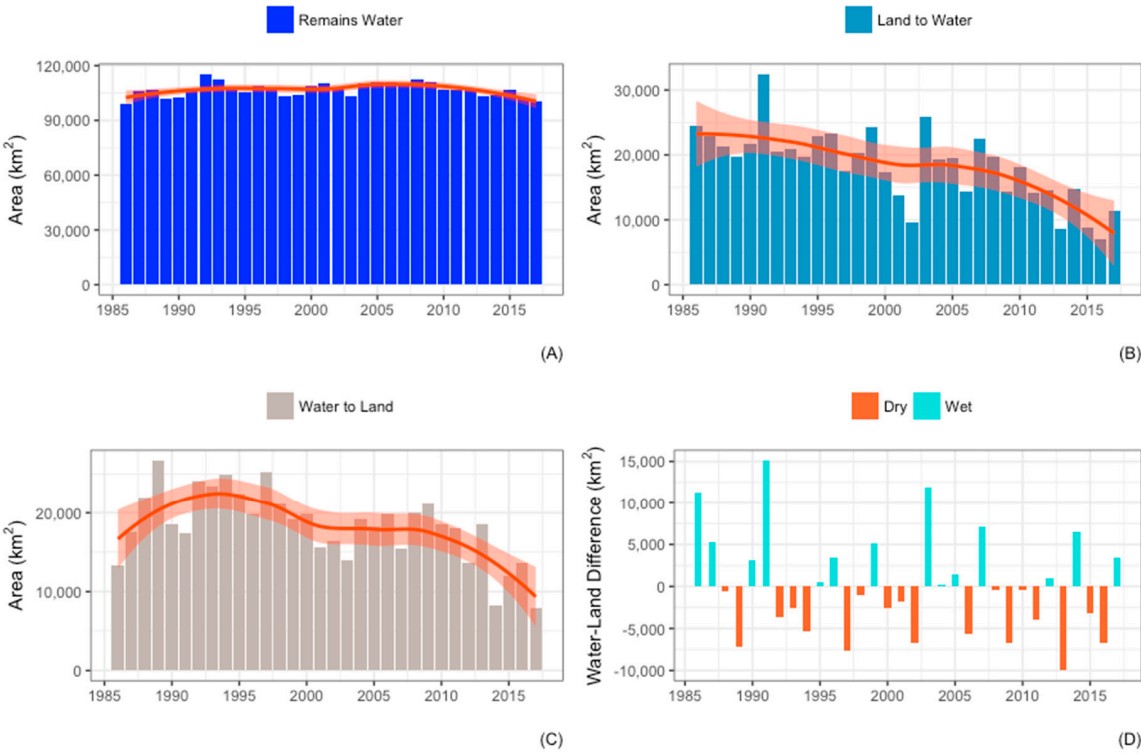

**Figure 5.** Annual decoupling of annual surface area mapped in the class Remains Water (**A**) from changes from Land to Water (**B**), Water to Land (**C**), and annual net difference (**D**) between the classes in (**B**,**C**).

The average change from Land to Water was 19,622 km$^2$/year over the 33-year timespan of this analysis (Figure 5B). We expected to detect the peak of this type of change in the years of flooding areas by hydroelectric dams inundation, but our time-series analysis did not reveal that signal because this analysis was conducted on the biome scale. More spatially detailed analysis (i.e., at the micro-watershed level) was also conducted and is presented in the following section. Flooding of natural landmasses in floodplains and along the borders of rivers and lakes makes are areas more subject to Land to Water changes. Moreover, extreme-flooding events were also not detected at the Amazon Biome scale. In 2012, for example, an extreme flooding event hit the port of the city of Manaus, increasing the water level by almost 30 m for more than two months [46,47], and an intense rainfall occurred in 2009 [48]. None of these extreme-flooding events are evident (Figure 4B) at the Biome scale. We have also observed that the average annual area that had interchange between water and land before 2010 (i.e., ~40,000 km$^2$) reduced by half in 2017 (Figure 5B,C). We did not observe opposite trends between Land to Water and Water to Land because these processes occur in different regions across the Amazon Biome.

The annual net difference of areas that interchanged between land and water may be more indicative of climatic extremes at the Biome scale (Figure 5D). This net difference can be an indicator of whether surface water expanded or reduced more in a given year, revealing an overall wet or dry condition in the Amazon region. The SWSC algorithm detected more surface water expansion in the 1980s, 1990s, and 2000s, and less in 2010s (Figure 5D). In 1995 and 2005, years reported as having a deficit in rainfall [14], we observed a small net increase in areas changing from Land to Water. However, in 1997–1998, 2010 and 2016, years of extreme drought, we detected a net increase in areas changing from Water to Land.

We also analyzed the net difference of areas that interchanged between land and water between 1991 and 2017. These were the years that exhibited the highest and lowest surface water extent, respectively (Figure 4). The majority of the water to land change occurred in wetlands and flood plains along rivers and lakes (Figure 6). The interchange between water and land occurred, for example, with the construction of the Belo Monte hydroelectric dam (Figure 6), where gains in water occurred in the land area that was inundated, and losses in water happened in areas where the Belo Monte dam change their natural course (Figure 6). We conducted this same type of analysis using the GSW dataset and did not observe the changes presented in Figure 6 (see Figure S5).

*3.3. Characterization of Surface Water Types*

Surface water bodies were isolated and classified as Natural or Anthropic. The Anthropic surface water bodies were then sub-classified into Hydroelectric Dams, Agricultural Dams and Water in Mining (i.e., mostly gold mining operations). The Natural water bodies included large and small rivers, lakes and floodplains, but no attempt was made to separate these classes due to computational challenges in processing vector data in the Earth Engine.

Natural water bodies had an average surface area of 118,084 $km^2$/year (s = 7804) between 2000 and 2017, which is the period in which we analyzed the impact of deforestation on surface water. The statistical range for this class of surface water was 25,367 $km^2$ with the extremes in 2007 (125,845 $km^2$) and 2016 (100,478 $km^2$) (Figures 7 and 8A). The year 2016 is considered a "record-breaking warming" due to the extreme drought associated with El-Niño [49], and we found again a signal that shows a potential drought effect on the extent of surface water at Biome scale. We also found that the five lowest extents of Natural surface water happened between 2013 and 2017, the warmest decade in post-industrial history [14,50].

The area of Anthropic water bodies increased between 2000 and 2017 (Figures 7 and 8). We have identified three types for this class of water bodies: Hydroelectric Dams, Agriculture Dams and Water in Mining. The maximum extent of surface water for the Anthropic water bodies was detected in 2017, when it was 8831 $km^2$ (Figure 8), which represents a 28% increase relative to the average for this period (6899 $km^2$/year). The sub-classification of Anthropic water bodies showed that the area of Hydroelectric Dams was around 6000 $km^2$ between 2000 and 2010, but quickly increased after 2010, reaching 7467 $km^2$ in 2017 (i.e., ~85% of the surface area of Anthropic water bodies). However, Agriculture Dams have increased steadily between 2000 and 2017, from 457 to 1223 $km^2$, respectively (Figure 8C). The surface water extent of Agriculture Dams represents almost 14% of the total Anthropic water bodies in 2017, and this class spatially correlates with deforestation, with more than 50,000 small dams detected. The concentration of small agriculture dams also followed the spatial pattern of the main roads along the Transamazon and Santarém-Cuiaba highways (Figure 7).

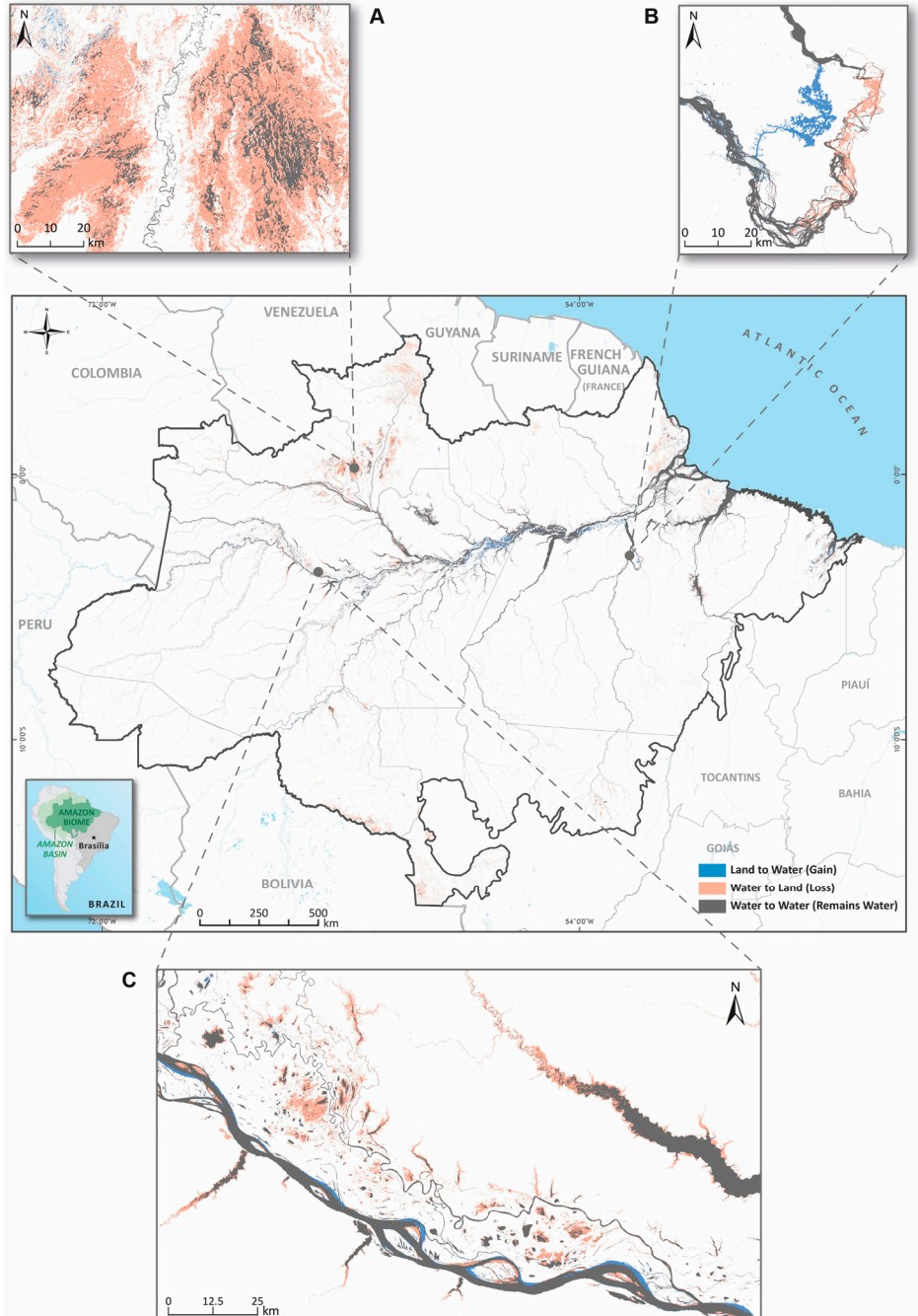

**Figure 6.** Interchange between water and land detected between 1991 and 2017, which are the highest and lowest surface water extent, respectively. Orange areas represent areas that changed from water in 1991 to land in 2017, and blue ones are changes from land to water between these years. In panel (**A**) wetlands changed from Water to Land; in (**B**) Land to Water associated with the construction of Belo Monte hydroelectric dam and Water to Land along the rivers that had the water flow diverted by the dam construction; and in (**C**) mostly Water to Land in flood plains along lakes and rivers.

The map of Anthropic water bodies revealed a spatial pattern that links fragmentation of small rivers with the Amazon Arc of Deforestation (Figure 7). Hydroelectric Dams are a well-known driver of fragmentation of large rivers dramatically affecting freshwater connectivity [9,51]. Fragmentation of small streams has not been documented yet at a large scale such as in the Amazon Biome. A study at the local level of the Paragominas municipality, in the Eastern Amazon, revealed high rates of deforestation in riparian permanent preservation areas, which are meant to protect small rivers and maintain forest

and water connectivity [52]. Our results showed that deforestation in riparian permanent preservation areas is happening in a much larger area across the Amazon Arc of Deforestation due to large number of dams along small streams (Figure 7).

Finally, we detected only 140 km² of surface water in the Water in Mining class, which represents 1.5% of the total Anthropic water bodies mapped until 2017 (Figure 8C). This class is mostly concentrated mostly in the Tapajós watershed in Southwestern Pará, but there are scattered spots of Water in Mining in Roraima, Rondonia and Mato Grosso states as well (Figures 7 and 8C).

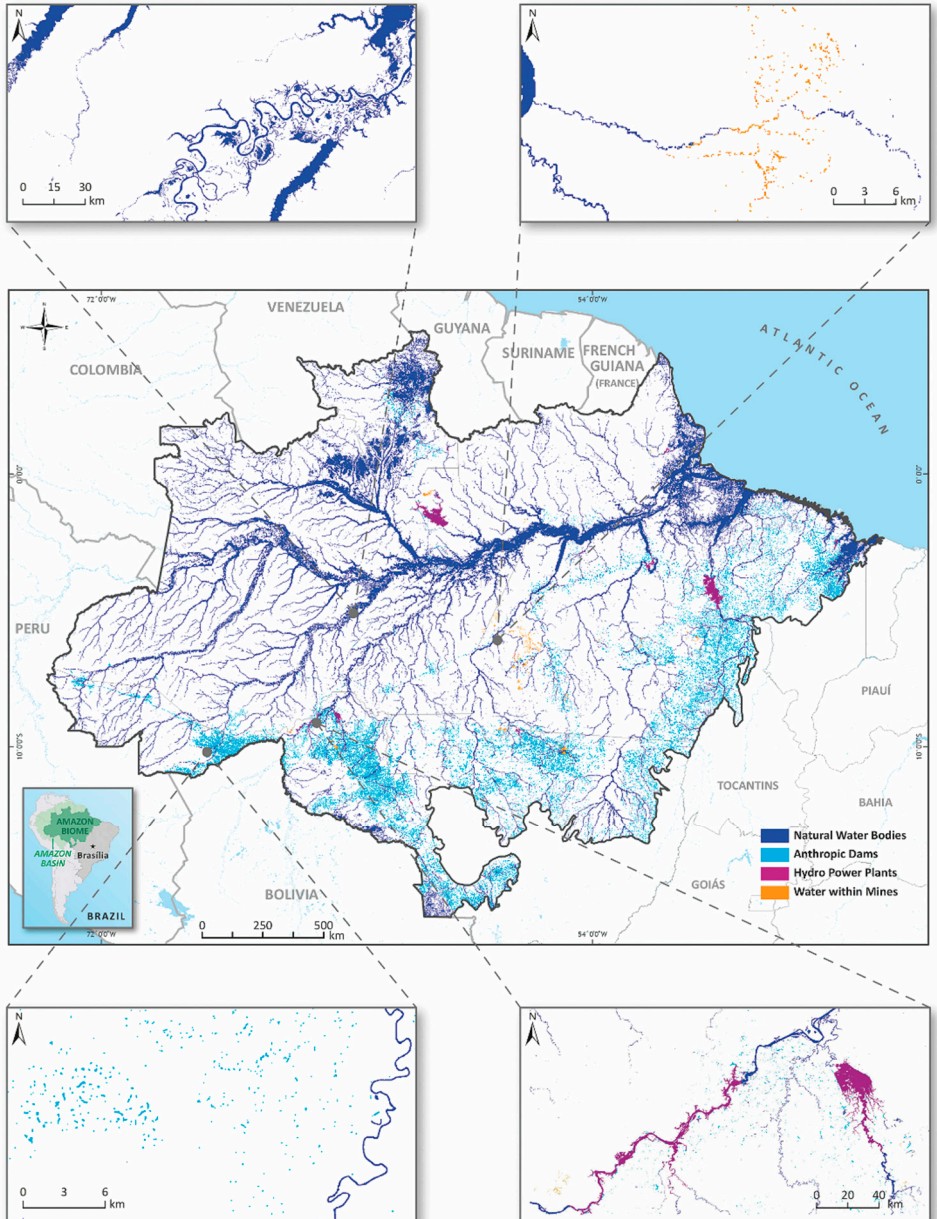

**Figure 7.** Surface water types mapped for the entire Amazon Biome in 2007. The panels depict examples of hydroelectric dams (bottom-right), small stream fragmentation in the Arc of Deforestation (bottom-left), Natural surface water (top-left) and mining along rivers (top-right).

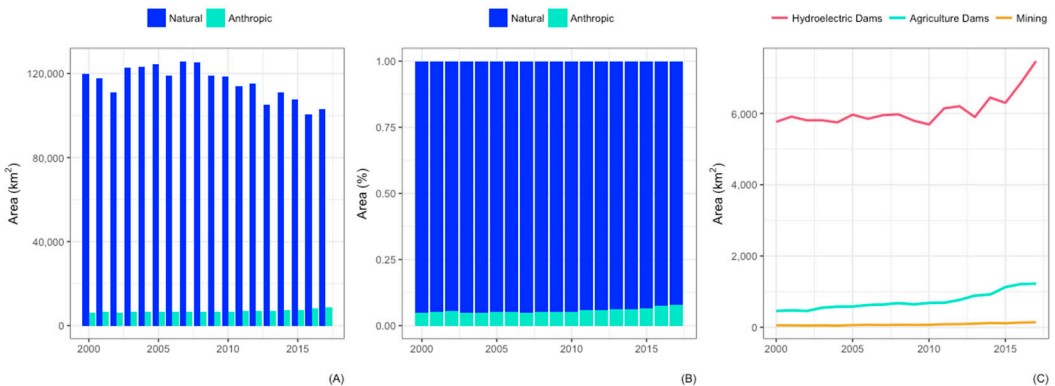

**Figure 8.** Absolute (**A**) and relative (**B**) surface water types showing a decrease in Natural water bodies, and an increase in Anthropic ones. Hydroelectric Dams showed an increase in surface water after 2010, and Agriculture Dams steady growth since 2000 (**C**).

### 3.4. Potential Drivers of Surface Water Change

The surface water dynamics at the watershed scale showed a trend of decreasing surface water for most watersheds, with some showing an increase trend (Figure 9). Figure 9A shows the proportion of deforestation within watersheds, with green areas representing watersheds with high forest cover and low level of deforestation, and yellow-red colors high deforestation and low forest cover (Figure 9A). The percent of losses and gains from the average surface water during 1985 and 2017 is shown for 2017 (Figure 9B). These maps revealed that most of the watersheds had their surface water reduced in both high deforestation and high forest cover conditions (Figure 9), implying that there is no correlation between both (i.e., there is reduced surface water regardless of the forest cover and deforestation).

The statistical test reviewed that the effect of deforestation on surface water change was slightly positive, but was stronger and more statistically significant ($\chi^2_1 = 53.1$; $p = 0.0023$) for the watersheds that gained water than for watersheds that lost water (Table S1). The effect of forest cover was stronger and more statistically significant ($\chi^2_1 = 142.7$; $p < 0.00001$) than deforestation, suggesting that watersheds with higher forest cover have lost more surface water (Table S2).

These results suggest that watersheds that have high forest cover (i.e., with minimal influence of land use and land cover change from deforestation in infrastructure development) may have other factors driving the reduction of surface water. The GSW dataset was also analyzed to assess whether surface water was reducing over time, but this trend was not detected. First, GSW detected also more surface water extent in the dry season relative to the wet season. A notched boxplot revealed that detection of surface water in these seasons was statistically different (Figure S3). We also observed that GSW did not detect land and water interchange as SWSC detected (Figure 6; see Figure S5 for comparison). Most of these changes are associated with wetlands, floodplains, and small streams, which are mixed water ecosystems composed of water vegetation and soils at the Landsat pixel scale. This may explain why GSW has not detected the same trend of reducing surface water after 2010 as detected by SWSC because GSW does not detect information at the sub-pixel level. Monitoring wetlands, floodplains and small streams are important because they are more vulnerable to climate change [53]. Further investigation is necessary to assess whether climate change can explain this overall trend of decreasing surface water after 2010 considering external drivers such as El Niño and the warming of the Atlantic North (Figure 4A).

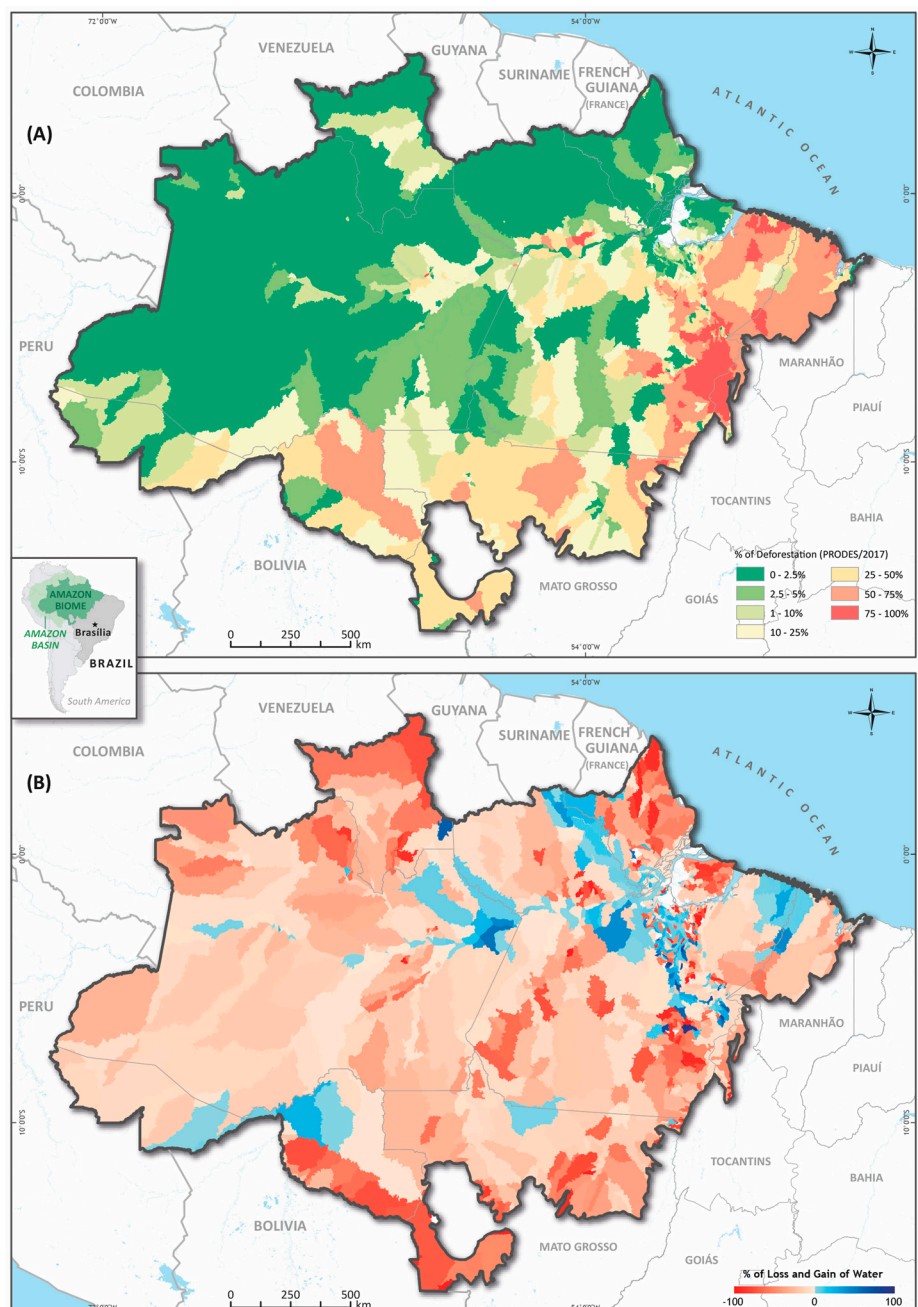

**Figure 9.** Proportion of forest (green) and deforested areas (orange) at level 4 watersheds (**A**), and gains and losses of surface water in 2017 per watershed relative to the mean difference from 1985 to 2017 (**B**).

## 4. Conclusions

We used Landsat images acquired during June and October for the years 1985 through 2017 to map and estimate the annual extent of surface water across the Amazon Biome in Brazil with the novel SWSC. This classifier has as inputs information about water (as represented by shade), green vegetation and soil endmembers found at the Landsat sup-pixel scale. The SWCS allowed detection and mapping of large water bodies (i.e., river and lakes) also detected with whole pixel classifiers such as GSW. However, the SWSC identified more surface water in wetlands, floodplains along rivers and lakes, and in narrow streams, and also allowed monitoring annual interchange between water and land. These regions with high water dynamics are even more vulnerable to climate change (e.g., [53,54]).

The SWSC used a generic SMA model using an endmember library extracted from Landsat datasets in a large time-series dataset. Therefore, this method had the potential to be applied at a global

scale. However, water in wet forests cannot be detected, which limits the use of optical imagery to map and monitor the full extent of surface water in the Amazon Biome. Landsat data also cannot monitor this region over the entire year, being more suitable to be used in the dry season when clouds are less abundant and frequent. The SWSC may be limited to monitor surface water in mountainous areas since shadows can obscure and create ambiguity with surface water. Our future research will integrate Sentinel 1 microwave data with optical data to improve monthly mapping and monitoring of surface water. These types of application will become more relevant, given the ongoing extreme warming and flooding events in this region.

The SWCS mapped an average of 130,000 km$^2$/year of surface water and detected a general trend of reducing surface water after 2010, the warmest years in this region since the post-industrial era in the Amazon [55]. This temporal trend of decreasing surface water was also visible in areas undergoing more annual interchange between landmass and water. Natural surface water bodies also had a decrease in extent over time. This process may be associated with a warming temperature trend in the Amazon region [55], also revealed in more detailed analysis at the watershed scale. The fact that watershed with high forest cover lost surface water over time may suggest that this process is connected with climate change. However, other studies have found a general wetting trend in the Amazon Basin [54,56], but our results suggest that during the dry season a drier trend has occurred in this region since 2010.

Most of the surface water mapped in this study was associated with large rivers, lakes and floodplains. Hydroelectric dams were found to have the largest area of surface water, with an increase after 2010 due to new constructions. Small narrow streams were not visible in forested regions but became distinct in the Arc of Deforestation where we detected more than 50,000 small dams between 2000 and 2017. Studies on the impact of land cover change on freshwater ecosystems have not addressed the direct disruption of small streams [2,57] by the construction of these small dams that serve mostly cattle ranching activity, followed by aquaculture and mining to a lesser extent. A local study showed that deforestation is happening in riparian protected reserves [52], fragmenting small streams. However, this study has demonstrated that this process is happening on the scale of the entire Amazon Biome.

The results presented here reinforce the idea that land management policies, which integrate terrestrial and freshwater issues, are crucial to the longevity of this unique freshwater system. Additionally, the results represent an important source of information to develop water management strategies capable of facing the challenges of climate change, land use and infrastructure. Finally, it can serve as a basis for the establishment of a national monitoring system that can provide high-value information to inform decision-making in the Amazon.

**Supplementary Materials:** The following are available online at http://www.mdpi.com/2073-4441/11/3/566/s1, Figure S1. Visualization and download interface on Google Earth Engine. Google Earth Engine Link to run the app above (copy and paste it to Code Editor and run it): https://code.earthengine.google.com/3a4e78799b0726409de84dc80b572bd8. Figure S2. Correlation matrix of all variables in the models described above. Figure S3. Boxplot of the monthly GSW detection of surface water between 1987 and 2015 showing more surface water detected in Dry months, relative to Wet months. Figure S4. First surface water occurrence in the GSW (A) and SWSC (B) between 1985 and 2015. Figure S5. Interchange between water and land detected between 1991 and 2017 obtained with GSW dataset for the Dry season. These years showed the highest and lowest surface water extent, respectively, using SWSC (see Figure 6 in the manuscript for comparison). Table S1. Coefficients for Model 1—watersheds that have annually gained surface water. Table S2. Coefficients for Model 1—watersheds that have annually lost surface water.

**Author Contributions:** C.M.S.J. Conceptualization, writing—original draft preparation, methodology, writing—review and editing; F.T.K. Software and data curation; M.H.S. Conceptualization, validation; J.G.R. Data curation, visualization; and B.C.O. Conceptualization, writing, review and editing, and funding acquisition.

**Funding:** This research and the article processing charges was funded by WWF-Brasil and by the MapBiomas Project.

**Acknowledgments:** We are grateful for the funding provided by the Norwegian Government to the MapBiomas Project via the World Resource Institute. We are also thankful to Google for providing access to the Earth Engine platform and for their technical support. Finally, we appreciate and thank the careful review and constructive critiques of three anonymous reviewers of this manuscript.

**Conflicts of Interest:** The authors declare no conflict of interest.

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
