# Peer review of "Long-Term Annual Surface Water Change in the Brazilian Amazon Biome: Potential Links with Deforestation, Infrastructure Development and Climate Change"

_water, doi:10.3390/w11030566_

Round 1

Reviewer 1 Report

Souza Jr et al. use Landsat and PRODES data to analyze the extent of surface water throughout the Amazon between 1985 and 2017 (but potentially 2015?). They used Google Earth Engine to perform a computationally intensive remote-sensing work to determine surface water extent throughout the Amazon. The author’s find that that there is an overall decreasing trend in surface water and link this to deforestation and hydroelectric dams post 2010. While the research design seems great and the results will be important to community, much of the methodology can be expanded upon and clarified, and some of the results were muddled. Improving these and the Figures will greatly improve the paper.

Major Edits

Line 61: Upon trying to look up citation 25 – as I was extremely interested in the methodology of the SWSC algorithm, I could not find this reference and the whole paper is based upon it.

Line 76: Because Google Earth Engine is an open-source platform, one could create the sharable link to be published with the text and provide the code to readers who want to replicate this analysis, or adapt it for other regions or ecosystems. This sharing of methodologies would be a HUGE contribution of this paper.

Lines 174-178. What metrics were used to differentiate between Natural (rivers/lakes/wetlands) and Anthropic (dams, mining) surface water?

Lines 187. Relabel this section “Results and Discussion” and the final section “Conclusions” OR present only the Results in this section with no commentary on the findings themselves.

Lines 189-195. This paragraph belongs in your methods section when the dates used for the algorithm are described, not the results section.

Figures throughout: They should not simply be screenshots of the Google Earth Engine interface – real map figures must be created.

Line 224 and throughout: What is “s” – error? Or standard deviation? If error – how was it calculated?

Section 3.2 was hard to follow and I got lost with the results. I am not sure how conclusions were reached from this section.

Lines 284-288. How were these Natural and Anthropic surface water features then sub-classified – explain in methods. What characteristics / cut-offs were used?

Lines 334-347. This paragraph just reiterates what was said in the previous section. Can remove entirely.  

Minor Edits

Line 42: Citation 12 – find a peer-reviewed source for this statistic

Line 43 and throughout: ‘water ecosystems’ is an awkward turn of phrase

Line 72: Spell out what PRODES data is. 

Section 2.1: Lines 83-91 are redundant from the intro

Lines 96/104. Does this study cover through to 2015 or 2017?

Line 254: Aren’t these changes annual? Not biannual? Again, a tightening of the writing throughout this paper would improve presentation of the interesting findings. 

Author Response

See uploaded file with the response.

Reviewer 2 Report

Dear Authors,

I've had a pleasure reading your manuscript, it is very interesting to see more research focusing on surface water changes focusing on a detailed analysis of these changes and possible causes as well as anthropogenic factors. Classification of surface water into natural and anthropogenic water types is a very important step to better understand surface water dynamics locally and globally. The paper presents an interesting method to classify surface water at the sub-pixel scale. 

In my opinion, the paper needs to be improved in a few areas to make the results of the research more convincing:

1. The resolution of figures is extremely low, and also formatting is far from perfect - please reformat all figures containing maps so that they will include lat/lon coordinates. Avoid compressing images as JPG. Consider exporting images and composing them using QGIS or at least add lat/lon coordinates, frame, scalebar as in https://code.earthengine.google.com/?accept_repo=users/gena/packages&scriptPath=users/gena/packages:style-test-frame-scalebar. And don't compress images as JPG, it makes very hard to see details. The use of higher DPI would be preferable as well. Also, Google Map text seems to be in German, I'm not sure this is consistent with the rest of the text.

2. It might be good to reflect the actual major changes identified during the research in the Abstract (the actual surface water extent changed, as discussed in 3.2

3. I really miss a map showing the (upscaled) changes between e.g. 1991 and 2015 in addition to the results presented in Figure 4 and 5. This would show where the main changes have occurred. Maybe also indicating locations of main changes as presented in Figure 2 of 23.

4. It is interesting to see that the results presented in the paper are contradictory (at least at the first look) to the [23, 24]. I did not analyze the whole study area in details, but it seems that the main Amazon River got wetter (at least in some areas) during the specified period. See figures1.zip, indicating changes near Urucurituba:

* Aqua Monitor: Amazon_AquaMonitor.png, Amazon_AquaMonitor_composite_2000.png, Amazon_AquaMonitor_composite_2011.png

I've adjusted DOY to match the ones used in the paper, but it still shows an increase of the surface water extent. By default, it uses 20% percentile by default to computing TOA reflectance composite images.

http://aqua-monitor.appspot.com/?mode=dynamic&from=2000&to=2011&view=-2.6500002200144,-57.426795412219356,11z&min_doy=150&max_doy=300. 

* JRC: Amazon_JRC.png

* Google Earth Engine Timelapse video: https://earthengine.google.com/timelapse/#v=-2.57229,-57.35512,10.147,latLng&t=3.20

Of course, this is a very small section along 3. and both of these examples use a relatively long time interval to estimate surface water changes.

Please discuss these differences, are your results different because [23,24] use different temporal baselines? Using JRC monthly dataset it should be possible to compute surface water extent trend very easily, maybe it would be a good idea to add it to Figure 4, I'd expect the surface extent to be lower than the one studied in your research and/or add an example showing area where your results contradict existing research results?

5. Did you consider sharing at least image assets of the results as a part of the paper? It is trivial to do with the Earth Engine and would dramatically improve the readability of the paper. Maybe making a very simple APP showing your results using https://developers.google.com/earth-engine/apps would be an option? Or at least as an EE script link presenting major map layers (water occurrence, changes, typology, etc.).

6. [25] is formatted incorrectly (no year, issue number, etc.), I could not find that paper and it is mentioned as the major reference explaining the SWSC method.

7. line 118. - why 500m, is this applied only for L5 and L7? I can imagine applying this to make SLC-OFF errors smaller.

8. line 184. - the link "here" is not a proper citation

Sincerely yours,

Reviewer

Author Response

See the uploaded file with the response.

Reviewer 3 Report

Reviewer Comments on manuscript submitted for publication Water-MDPI

Manuscript: Long-term annual water change in the Brazilian Amazon Biome: links with deforestation, infrastructure development and climate change

Souza, et al.,

               The main goal of the manuscript is to produce an annual surface water map of the Brazilian Amazon biome in the dry season. As sub products the authors analyze total water area of the region as well as by different water bodies (natural, anthropogenic and sub-groups) and quantify their changes in time and qualitatively address the main possible drivers for the changes. The work in general is relevant, and presented intensive methodological computational skills, is well written and a clear structure. In general I would recommend a detailed description of the novelty, relevance and limitation of the results and the product generated to future researchers.

I recommend that this manuscript goes through major revision, before being considered for publication. Comments and recommended changes are listed below.

1)      The Introduction well describes the problems and motivations, the tools used, and what they present as results in this manuscript. But it doesn’t give a background of any product of that kind in the region, so it would be good for the manuscript to make it clear how novel the study is for the amazon region, if not what are the references and how their results compares to previous studies.

2)      The manuscript provides two main aspects: the product of annual map of surface water in the Brazilian Amazon and the analyses, quantification of changes and attribution. How does the product compares to Pekel, 2016Nature or others, if any.  It would be important for the reader also to understand how novel the quantification of water body in the Amazon region is, or how it compares to previous estimates. How this product can be used, and how it can address the motivations in the first paragraph of introduction, and also what are the limitations of the product for the different applications.  This is an important product and it doesn’t mention in the text if it will be available and where.

2.1) For example the water surface product reflects only the dry season, and optical Satellite as Landsat are only able to capture open water information, missing water surfaces under tree canopy. These aspects are listed across the manuscript but would be useful to have a summary of the uses and the limitations of it in one place.

3)      The analyses of the entire Brazilian biome as a whole (Figure 4, Table1) present a strong signal of change. Despite of the limitations of the product (cited before) the changes are relative to the initial conditions, and seems very relevant, a strong result of the manuscript.

4)      The attempt to relate climatic events to the biannual shifts between Land-Water and Water-Land seems to be limited due to the limited window of dry season. The flood events are probably not noticeable in this study because of that. This attempt seems also to be limited due to the large scale of the analyses, once the events of El Nino or North Atlantic oscillation affect different location and might even present opposite effects within the studied region, and also limited to the dry season timeframe. An analyses of this effect would be valid within watersheds, where the water balance could be fully done. The analyses of Figure 5 should be revisited.

5)      It was not clear how deforestation data was related to shifts in water surface areas.

Author Response

See the uploaded file with the responses.

Round 2

Reviewer 2 Report

I'm satisfied with the answers.

Author Response

Thanks for the careful review and constructive feedback.

Reviewer 3 Report

Reviewer Comments on manuscript submitted for publication Water-MDPI

Manuscript: Long-term annual water change in the Brazilian Amazon Biome: links with deforestation, infrastructure development and climate change 

Souza, et al., 

The main goal of the manuscript is to produce an annual surface water map of the Brazilian Amazon biome in the dry season. As sub products the authors analyze total water area of the region as well as by different water bodies (natural, anthropogenic and subgroups) and quantify their changes in time and qualitatively address the main possible drivers for the changes. The work in general is relevant, and presented intensive methodological computational skills, is well written and a clear structure. 

The authors have satisfactory worked on the major revision, the addition of the watershed level analysis was a significant improvement in the manuscript results, I recommend it for publication.

Author Response

(The authors gave the same response as above.)
